# Eating Quality of Pork from Entire Male Pigs after Dietary Supplementation with Hydrolysable Tannins

**DOI:** 10.3390/ani13050893

**Published:** 2023-03-01

**Authors:** Ivan Bahelka, Roman Stupka, Jaroslav Čítek, Michal Šprysl, Ondřej Bučko, Pavel Fľak

**Affiliations:** 1Faculty of Agrobiology, Food and Natural Resources, Czech University of Life Sciences Prague, 165 00 Prague, Czech Republic; 2Faculty of Agrobiology and Food Resources, Slovak Agricultural University Nitra, 949 01 Nitra, Slovakia

**Keywords:** sensory traits, boars, diet, hydrolysable tannins, consumers

## Abstract

**Simple Summary:**

The European pig industry needs to adapt to growing social interest regarding animal welfare. One of these concerns is the surgical castration of male piglets—a common practice primarily performed to avoid the risk of boar taint released from the meat of uncastrated males, especially during heat treatment. Several EU countries are trying to stop the surgical castration of pigs. The European Commission (EC) strongly supports these activities. One of the two currently feasible alternatives to the production of castrates is the fattening of entire male pigs. It is well known that skatole, one of two main compounds responsible for boar taint, can be eliminated or reduced by feeding additives. Recently, some promising results have been achieved using hydrolysable tannins in the diet of entire males. However, it should be mentioned that these studies focused on the influence of tannins on fattening, carcass value, meat quality and the deposition of androstenone and skatole in adipose tissue but not their influence on sensory characteristics. Therefore, the objective of this study was, in addition to determining the effects of tannins on skatole and androstenone accumulation in fatty tissue, to assess the sensory attributes of pork from entire males after supplementation of the diet with 1–4% tannins. The results showed that 2–4% supplementation of tannins in the feed reduced the accumulation of skatole in fatty tissue. The odour and flavour of pork were not influenced by tannin supplementation, but higher doses of tannins decreased the juiciness and tenderness of pork from entire males but only in men’s evaluation. The effect of the sex of the panellists on both of these sensory traits was observed in both the control and tannin-supplemented groups.

**Abstract:**

Boar taint is an unpleasant odour and flavour released during heat treatment of pork from uncastrated male pigs. The two main compounds responsible for boar taint are androstenone and skatole. Androstenone is a steroid hormone formed in the testis during sexual maturity. Skatole is a product of microbial degradation of the amino acid tryptophan in the hindgut of pigs. Both of these compounds are lipophilic, which means that they can be deposited in adipose tissue. Several studies have reported heritability estimates for their deposition from medium (skatole) to high magnitudes (androstenone). In addition to efforts to influence boar taint through genetic selection, much attention has also been paid to reducing its incidence using various feeding strategies. From this point of view, research has focused especially on the reduction in skatole content by supplementation of feed additives into the nutrition of entire male pigs. Promising results have been achieved using hydrolysable tannins in the diet. To date, most studies have investigated the effects of tannins on the production and accumulation of skatole in adipose tissue, intestinal microbiota, growth rate, carcasses and pork quality. Thus, the objective of this study was, in addition to determining the effects of tannins on androstenone and skatole accumulation, to assess the effects of tannins on the sensory traits of meat from entire males. The experiment was performed on 80 young boars—progeny of several hybrid sire lines. Animals were randomly assigned to one control and four experimental groups (each numbering 16). The control group (T0) received a standard diet without any tannin supplementation. Experimental groups were supplemented with 1% (T1), 2% (T2), 3% (T3) or 4% (T4) SCWE (sweet chestnut wood extract) rich in hydrolysable tannins (Farmatan). Pigs received this supplement for 40 days prior to slaughter. Subsequently, the pigs were slaughtered, and sensory analysis was applied to evaluate the odour, flavour, tenderness and juiciness of the pork. The results showed a significant effect of tannins on skatole accumulation in adipose tissue (*p* = 0.052–0.055). The odour and flavour of the pork were not affected by tannins. However, juiciness and tenderness were reduced by higher tannin supplementation (T3–T4) compared to the controls (*p* < 0.05), but these results were sex-dependent (in favour of men compared to women). Generally, women rated tenderness and juiciness worse than men regardless of the type of diet.

## 1. Introduction

Animal welfare has become a very important factor in livestock breeding, including that of pigs. A general ban on surgical castration of entire male pigs was expected in all EU member states by the end of 2018. Due to various circumstances, it was postponed until after 2021. One feasible alternative to painful surgical castration is to fatten all males. It is well known that boars grow faster and more efficiently, and they have higher lean meat content in carcass than surgical castrates. On the other hand, fattening boars increases the risk of and results in a higher incidence of boar taint and thus the dissatisfaction of consumers with such meat.

The incidence of boar taint is mainly attributed to two substances, androstenone [1] and skatole [2], especially after heat treatment of pork. Androstenone (α-androst-16-en-3-one) is a steroid hormone synthesized in the Leydig cells of the testis of uncastrated boars. This compound has an odour similar to urine or sweat, is perceived by approximately two-thirds of the human population and has been shown to be different according to country/region. Deposition of androstenone in fat tissue has high heritability estimates (0.55–0.88), which indicates that it is affected mainly by genetics [3,4]. Skatole (3-methyl-indole) is a metabolite derived from microbial catabolism of the amino acid tryptophan in the hindgut of pigs. Its deposition is influenced mainly by environmental factors (*h*^2^ = 0.23–0.55) [5], especially nutrition, the system of feeding and the housing conditions [6,7,8,9,10]. Since both of these compounds are lipophilic, they can accumulate in adipose tissue and therefore may have a negative effect on sensory attributes and result in the rejection of such meat by consumers.

Since the odour of skatole is negatively perceived by practically the entire human population, reduction efforts have focused mainly on various feeding strategies in fattening all males. Promising results have been achieved by supplementation of diets with a variety of feed additives, such as chicory root or inulin [11,12,13,14,15,16,17,18,19,20], raw potato starch [21,22,23,24], sugar beet pulp [25], Jerusalem artichoke [26], oligofructose and fructooligosaccharides [15,27] and tannins [28,29,30].

Tannins are plant metabolites with great diversity, resulting in different physiological effects according to their form, animal species and amount of supplementation [29,31]. Sweet chestnut (*Castanea sativa* Mill.) wood extract, consisting mainly of hydrolysable tannins, has antimicrobial and antiviral properties. Therefore, products containing this substance are used in animal nutrition, especially in piglets, as supportive treatment for diarrhoea after weaning [32,33,34,35,36,37,38].

Several studies have demonstrated a reduction in total protein digestibility, as well as inhibition of microbial activity, in the colons of pigs after supplementation of the pig diet with hydrolysable tannins [39,40,41]. Lower disponibility of tryptophan and cell debris in the hindgut may lead to reduced intestinal production of skatole [28,42]. These findings are interesting from the entire male production point of view.

Apart from a few studies dealing with Iberian pigs fed natural sources [43,44,45,46], other research on the effects of hydrolysable tannins (HTs) has been mainly aimed at growth and carcass parameters, meat quality and oxidative stability or the intestinal skatole production and microbiota composition in the large intestines of boars [28,29,30,31,32,37,41,47], but almost no research has addressed the effect of HTs on the sensory traits of entire male meat.

The main objective of the present study was to assess the impact of hydrolysable tannins on parameters of eating quality, considering the possible effect of the sex of consumers, as well as to investigate the relationships between sensory evaluation and the content of skatole and androstenone in adipose tissue.

## 2. Materials and Methods

### 2.1. Animals and Diet

Eighty young boars were used in the experiment. They were the progeny of Landrace sows and Yorkshire × Pietrain boars. Two weeks before the experiment, the pigs were housed in pairs/pens at the experimental test station of the Research Institute for Animal Production (RIAP) Nitra. Subsequently, the boars were randomly distributed within litters to one control and four experimental groups (each containing 16 animals). Control pigs (T0) received a diet without any supplementation. Experimental groups received the same diet as the control group but supplemented with 10 (T1), 20 (T2), 30 (T3) or 40 (T4) g/kg sweet chestnut wood extract (SCWE) rich in hydrolysable tannins (Table 1). The producer of the Farmatan product is Tanin Sevnica d.d., Sevnica, Slovenia, and the supplier was Product Feed a.s., Luzianky, Slovakia. The content of tannins in this product is 73 ± 2% (the value declared by the producer). The analysis of feed was performed according to the Folin–Ciocalteu method [48]. The total phenolic content is expressed as gallic acid equivalents and is 45.1%.

Supplementation of the diet with tannins started when the boars reached an average live weight of 80 kg (after a 2-week transitional period) and lasted for 40 days prior to slaughter. Access of the animals to drinking water and feed (automatic feeders—Schauer s.r.o., Nitra, Slovakia) was ad libitum.

### 2.2. Slaughter and Sample Collection

Entire males were slaughtered in one batch at the experimental slaughterhouse of RIAP Nitra. The average live weight of the pigs was 122.28 kg ± 5.63 kg. Standard slaughter conditions were used: electrical stunning (90–100 V, 0.9–1.0 A, 50 Hz) followed by exsanguination. Evisceration was completed approximately 20 min post mortem. Chilling of the carcasses (air temperature 2–4 °C, velocity 0.5–1.0 m.s^−^) started approximately 60 min after slaughter and was continued overnight. After 24 h of chilling of the carcasses, musculus longissimus thoracis (LT) samples (1.0 cm thick) with subcutaneous fat were removed from the right side of the carcass (at the level of the last rib) and stored at −20 °C until sensory evaluation.

### 2.3. Sample Preparation

One day before sensory evaluation, the raw LT sample was thawed overnight at 4 °C. Subsequently, each LT slice was trimmed to have 5 mm of subcutaneous fat. Each LT slice was placed in a grill and cooked for 4 min at 180 °C. The average measured internal temperature of the samples was 80 °C. After grilling, each steak was cut into four strips and immediately served to different panellists.

### 2.4. Sensory Evaluation

Panellists were recruited from the staff of RIAP Nitra. All of them were consumers who liked and ate pork regularly (2–3 times weekly). Before the sensory evaluation of samples, consumers were tested for their sensitivity to androstenone according to the modified method of Weiler et al. [49]. On the basis of this method, 20 panellists were selected (12 men and 8 women aged 32 to 60 years old).

In total, 320 samples were evaluated in eight sessions (each of 40). In each session, eight panellists evaluated five samples (one from each dietary group). Each panellist participated in several sessions. Meat samples were randomly served to the consumers, and the attributes classified were odour, flavour, tenderness and juiciness. The scale applied in the sensory test was structured into 5 points, with 1 being the worst and 5 the best evaluation (Table 2). The panellists were asked to score for odour first, followed in order by flavour, tenderness and juiciness.

### 2.5. Skatole and Androstenone Determination

Fat samples (100 g from a part of the belly) from entire males were removed 24 h after slaughter. One part of each sample was individually packed in a microtene bag, marked and frozen (−20 °C) until panel testing. The second part of each fat sample was transported to the authorized private laboratory of EKOLAB, s.r.o. (Košice, Slovakia), to analyse the androstenone and skatole concentrations according to the methods of Ampuero Kragten et al. [50] and Bekaert et al. [51]. Briefly, adipose tissue samples were melted in a microwave for 4 min. Liquid fat was transferred to centrifuge tubes (2 mL), and 1.75 mL of extraction solvent methanol:hexane (9:1) were added. The extract was ultrasonically cleaned in a bath at 50 °C for 5 min and centrifuged for 5 min at 10,000× *g*. After cooling, approximately 2.0 mL of extract were then injected into a gas chromatograph equipped with a mass spectrometry (MS) detector (Shimadzu GCMS-TQ8030, Shimadzu Corp., Kyoto, Japan) at an injection temperature of 260 °C.

The limits for detection were 0.02 µg/g for androstenone and 0.01 µg/g for skatole.

### 2.6. Statistical Analysis

The observed data were evaluated by 1-way analysis of variance (ANOVA) with fixed effects [52] using the following model:

y_ij_= μ + α_i_ + e_ij_ with N (0,σ^2^) for androstenone and skatole

For multiple comparisons of treatment means, Bonferroni’s test was used [53]. Dunnett’s test was not used to compare the control and treatment groups since, from the analyses of variance of both traits, it could be concluded that differences between all groups by the F test were nonsignificant. The observations of sensory traits were evaluated by 2-way ANOVA with fixed effects using the following statistical model:

y_ijk_ = μ + α_i_ + (αβ)_jj_ + e_ijk_ with N (0,σ^2^)

Pearson’s correlation coefficients of androstenone and skatole with sensory traits were calculated.

## 3. Results

### 3.1. Effect of Tannins on Androstenone and Skatole

Levels of boar taint compounds in adipose tissue with regard to tannin supplementation are shown in Figure 1. The samples that were high in both androstenone and skatole levels (greater than thresholds of 1.0 and 0.2 µg/kg, respectively) in the control group numbered 5, whereas in the supplemented groups, they numbered 1, 2, 3 and 0 for the T1, T2, T3 and T4 groups, representing 31.25%, 6.25%, 12.5%, 18.75% and 0%, respectively. The total number of samples high in androstenone, skatole or both compounds was 10 (62.5%). After tannin supplementation, these numbers were reduced to 7%, 6%, 8% and/or 7% (43.75, 37.5, 50.0 and/or 43.75%).

Deposition of androstenone in adipose tissues was not affected by supplementation of the EM diet with tannins (Table 3). A numerical reduction, however nonsignificant (*p* = 0.052–0.055), in skatole concentration in fat tissue was found after administration of 2%, 3% and 4% tannins compared to the control group.

Correlations between observed traits are presented in Table 4. Generally, all the relationships of concentrations of androstenone or skatole in adipose tissue with sensory traits were small and negative. Higher correlations were observed between skatole and eating quality parameters, with the exception of juiciness, which was more correlated with androstenone concentration.

### 3.2. Effect of Tannins on Eating Quality of Pork

Supplementation of the pig diet with tannins did not show any impact on the panellists’ scores for meat odour and flavour (Table 5 and Table 6). However, significant differences among the treatment groups were found for tenderness and juiciness (Table 7 and Table 8). Men scored both traits significantly better in meat from the control EM group than in meat from the T3–T4 or T4 group (3.05 vs. 2.74 and 2.81; 3.28 vs. 2.75, *p* < 0.05). Moreover, significant differences in the evaluation of these two traits were observed between the sexes of the panellists. Women were more critical than men regarding both tenderness and juiciness traits regardless of diet group (2.70 vs. 3.00, *p* < 0.05, and 2.65 vs. 2.99, *p* < 0.01, respectively).

## 4. Discussion

Generally, tannins are considered to be antinutrients, as they create compounds with other nutrients, such as proteins, minerals, or digestive enzymes, and therefore are capable of reducing feed palatability, feed intake and nitrogen digestibility [39,54,55]. However, pigs, wild as well as domestic, seem to be relatively resilient to the intake of feedstuffs with a high content of tannins without any negative consequences on performance or health status [56]. This resistance is associated with elevated synthesis of proline-rich proteins (PRPs) in the saliva, which bind tannins from feedstuffs and prevent intoxication of organisms with diets rich in hydrolysable tannins [57,58]. Moreover, recent studies have suggested that tannins have antimicrobial, anticancer, and antioxidant properties and can improve feed efficiency and reduce bacterial proteolytic reactions in piglets, thus protecting them from severe diarrhoea during weaning [32,33,34,35,36,37,59]. It is well known that wild pigs, as well as some special native breeds of domestic pigs (Iberian, Cinta, etc.), can eat foodstuffs rich in tannins (content: 4–7%); therefore, a dose of 40 g per kg of feed mixture was selected as the highest dosage in the present study.

It is well known that the level of skatole in fatty tissue is influenced by many processes, such as formation, absorption, metabolism and deposition. The main role is associated with the activities of digestive enzymes in the CYP450 family, such as 2E1, 2A19, 1A1, and 1A2, in the liver [6,28].

Generally, data relating the effect of tannins on skatole production and accumulation in pig adipose tissue are limited. Some studies have shown that tannins can inhibit proliferation and apoptosis in the caecum. This inhibition results in decreased skatole production in the large intestine due to the lower availability of cell debris from lower apoptosis and tryptophan [41]. In the present study, skatole accumulation in adipose tissue tended to decrease with increasing tannin supplementation. Similar numerical decreases were observed in other studies after 2–3% [28] or 3% [29] tannin supplementation, but surprisingly, lower supplementation (1.0 or 1.5%) resulted in higher skatole accumulation than in the control group [28,30]. Čandek-Potokar et al. [28] suggest that this result could be associated with lower activity of CYP2E1 and CYP2A19 enzymes, two major enzymes of the phase 1 metabolism of skatole in the liver. It should be mentioned that some of the above studies [29,30] used products with lower contents (only 50%) of hydrolysable tannins than our study.

Regarding the effect of tannins on androstenone, the present study showed that supplementation with these additives did not have any impact, even though the two highest doses had higher (although nonsignificant) androstenone concentrations in fat tissue than the control group. This result is partially in contrast with that of other studies [28,29,30] in which numerical reductions were found after supplementation of diet with 1–3, 3, and 3% hydrolysable tannin extracts. However, the pigs in the latter study [30] had low levels of androstenone overall.

Generally, the results regarding the effects of tannin supplementation on skatole and androstenone levels in back fat were not consistent and were highly dependent on the dose of tannin supplementation, often reporting curvilinear dependence. Therefore, research to determine the optimal dose of tannin supplementation for reducing boar taint is still needed. Moreover, the effects of hydrolysable tannins on androstenone are still unclear, and if any are confirmed, further studies will be needed to clarify the mechanism of action.

As previously mentioned, very few studies have been published thus far relating the effects of hydrolysable tannins on pigs. Those studies focused mainly on intestinal skatole production, growth rate, carcass and meat quality, and the intestinal microbiota [30,32,37,47,59], but almost none focused on the sensory properties of pork. Thus, our results related to the effects of tannins on eating quality are difficult to compare with those of other studies. Only one study in pigs [30] and one in sheep [60] investigated the impact of tannin supplementation on the sensory traits of meat. In the present study, the odour and flavour of boar meat were not affected by tannin supplementation. Panellists scored these two parameters in the control and supplemented groups at almost the same level, and the differences were not significant. Bee et al. [30] reported different findings. Panel members detected a stronger boar taint odour but not flavour in the meat of entire males supplemented with 1.5% tannins compared to the controls and group with 3% tannins. In lambs, Priolo et al. [60] observed lower sheep meat odour in meat from animals supplemented with tannins compared to those not supplemented. Contrary to odour and flavour, the other two sensory traits—tenderness and juiciness—in the present study showed significant differences. Panellists, but only men, scored (*p* < 0.05) tenderness better in the control group than in the two supplemented groups (T3–T4), and juiciness in the control group was evaluated better than in T4. This outcome is contrary to the results of a study [30] that reported no significant effect of tannin supplementation on juiciness and/or tenderness. Regarding differences between the sexes of panellists in the present study, women scored tenderness and juiciness worse than men in both the control and supplemented groups.

It seems that higher supplementary tannin doses (3–4%) in our study significantly reduced juiciness (T4) and tenderness (T3–T4) compared with the control group. However, this result may be sex-dependent.

## 5. Conclusions

Supplementation with tannins had no effect on androstenone accumulation in adipose tissue. In contrast, the concentration of skatole in fat tissue tended to decrease with increasing tannin supplementation. The odour and flavour of pork were not affected by tannin supplementation, but its juiciness and tenderness were lower after supplementation of the entire male diet with higher concentrations of tannins. The effect of the sex of consumers on sensory evaluation showed significant differences for the tenderness and juiciness of pork in favour of men. It seems that tannin concentrations greater than 3 in the diet negatively affect some sensory characteristics. These findings might be useful in solving appropriate feeding strategies for entire males to reduce boar taint, as well as to maintain good eating quality of their meat.

## Figures and Tables

**Figure 1 animals-13-00893-f001:**
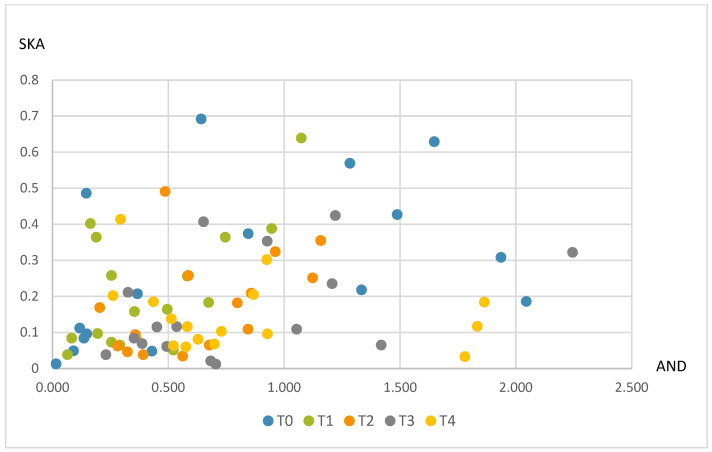
Diversity of androstenone (AND) and skatole (SKA) regarding tannin treatments.

**Table 1 animals-13-00893-t001:** Ingredients and chemical composition of diets.

Diet	T0	T1	T2	T3	T4
Ingredients, g/kg					
Wheat	150	150	150	150	150
Barley	360	360	360	360	360
Corn	150	150	150	150	150
Wheat bran	80	70	60	50	40
Soybean meal	110	110	110	110	110
Rapeseed meal	100	100	100	100	100
Hydrolysable tannins ^1^	0	10	20	30	40
Mineral suppl.	25	25	25	25	25
Premix	10	10	10	10	10
Ground limestone	10	10	10	10	10
Fodder salt	5	5	5	5	5
Analysis, g/kg					
DM ^2^	898.4	896.3	898.8	899.0	895.8
Crude protein	168.4	167.6	167.8	164.7	165.2
Crude fibre	50.8	51.6	49.9	51.4	50.5
Crude fat	27.2	26.5	25.5	26.4	26.7
Crude ash	43.7	45.8	44.9	43.3	44.7
ME ^3^, MJ/kg	13.9	13.8	13.8	13.7	13.8

^1^ SCWE—sweet chestnut wood extract, ^2^ DM—dry matter, ^3^ ME—metabolisable energy.

**Table 2 animals-13-00893-t002:** Scale for sensory evaluation of meat.

Trait	Scale				
	5	4	3	2	1
Odour	very distinctive, typical of roasted meat, without any foreign smell	distinctive, typical smell of roasted meat	less distinct, faintly typical aroma of roasted meat, with a faint foreign smell	atypical aroma of roasted meat with a stronger foreign smell	bland, impure scent with a strong foreign smell
Flavour	very distinctive, typical of roasted meat, without any extraneous flavour	distinctive, typical taste without noticeable foreign flavour	less distinct, faintly typical flavour of roasted meat, with a faint foreign flavour	bland and atypical taste of roast meat with a noticeable foreign flavour	bland with a foreign flavour, unpleasant to disgusting
Juiciness	meat very juicy	meat juicy	meat less juicy	meat almost dry	meat dry
Tenderness	meat slightly fibrous, very tender and soft	meat still tender, stringy, fragile and soft	meat thicker, fibrous, less fragile and quite firm	meat almost coarsely fibrous with stiff fibres, hard	meat coarsely fibrous, firm, very hard

**Table 3 animals-13-00893-t003:** Effect of tannin supplementation on androstenone and skatole accumulation in fatty tissue.

Trait	Treatment					SEM	*p* Value
	T0	T1	T2	T3	T4		
Androstenone, µg/g	0.79	0.43	0.62	0.80	0.84	0.064	n.s.
Skatole, µg/g	0.28 ^a^	0.22	0.17 ^b^	0.16 ^b^	0.15 ^b^	0.042	0.052–0.055

T0—control group, T1—1% tannin supplementation, T2—2% tannin supplementation, T3—3% tannin supplementation, T4—4% supplementation, ^a,b^ Different superscripts within rows indicate significant differences at *p* < 0.05; n.s.—non significant.

**Table 4 animals-13-00893-t004:** Pearson’s correlation coefficients between traits.

Trait	Skatole	Odour	Flavour	Juiciness	Tenderness
Androsterone	0.32	−0.02	−0.10	−0.19	−0.13
Skatole	-	−0.12	−0.13	−0.03	−0.17

**Table 5 animals-13-00893-t005:** Effect of tannin supplementation on sensory evaluation of odour.

	T0–T4	T0	T1	T2	T3	T4	SEM	*p* Value
Men	3.28	3.30	2.96	3.37	3.62	3.16	0.07	n.s.
Women	3.31	3.27	3.60	3.20	2.90	3.60	0.14	n.s.
Total	3.30	3.28	3.28	3.29	3.26	3.38	0.17	n.s.

n.s.—non significant.

**Table 6 animals-13-00893-t006:** Effect of tannin supplementation on sensory evaluation of flavour.

	T0–T4	T0	T1	T2	T3	T4	SEM	*p* Value
Men	3.32	3.39	3.13	3.43	3.43	3.21	0.06	n.s.
Women	3.25	3.37	3.60	3.13	3.13	3.02	0.11	n.s.
Total	3.28	3.38	3.37	3.28	3.28	3.11	0.13	n.s.

n.s.—non significant.

**Table 7 animals-13-00893-t007:** Effect of tannin supplementation on sensory evaluation of tenderness.

	T0–T4	T0	T1	T2	T3	T4	SEM
Men	3.00 ^1^	3.05 ^a^	3.40	3.00	2.74 ^b^	2.81 ^b^	0.06
Women	2.70 ^2^	2.87	3.00	2.20	2.63	2.81	0.11
Total	2.85	2.96	3.20	2.60	2.68	2.81	0.12

^a,b^ Different superscripts within rows indicate significant differences at *p* < 0.05; ^1,2^ Different superscripts within columns indicate significant differences at *p* < 0.05.

**Table 8 animals-13-00893-t008:** Effect of tannin supplementation on sensory evaluation of juiciness.

	T0–T4	T0	T1	T2	T3	T4	SEM
Men	2.99 ^‡^	3.28 ^a^	3.07	3.06	2.82	2.75 ^b^	0.05
Women	2.65 ^ʢ^	2.53	2.60	2.67	2.88	2.58	0.10
Total	2.82	2.91	2.83	2.86	2.85	2.66	0.12

^a,b^ Different superscripts within rows indicate significant differences at *p* < 0.05; ^‡, ʢ^ Different superscripts within columns indicate significant differences at *p* < 0.01.

## Data Availability

The data presented in this study are available upon request from the corresponding author.

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
