# Peer review of "Eating Quality of Pork from Entire Male Pigs after Dietary Supplementation with Hydrolysable Tannins"

_animals, 2023, doi:10.3390/ani13050893_

Round 1

Reviewer 1 Report

Dear authors,

This is an interesting and valuable paper.

A main general remark concerns the mention of tannins feed-supplementation. A confusion is generally made between the amount (in %) of supplemented extract, containing tannins, and the actual % of tannins supplemented. It is important to take into account this difference, especially when comparing results from other studies. Indeed, in the references mentioned here, some extracts contain 75% of hydrolysable tannins whereas others contain only 50%. Accordingly, it its also important to describe the method of determination of tannins.

Furthermore, it is important to clearly state when an effect is found its significance level. And even more, to clearly state for a numerical effect  found that it was not significant.

Specific remarks:

L 11:    Please insert “the risk of” after “to avoid”

L 19:    Please consider changing to “2-4% supplementation of tannins in the feed”

L57-58: Anosmia to Androstenone was shown to be different according to country/region!

L 104-105: “75%” is this figure the hydrolysable tannins content certified by the provider of chestnut extract? Or is it a value determined by you? In any case, please provide the method of analysis.

L 140-145: Please provide a short method description of this analysis and a reference. This is very important as it is the source of your main results together with sensory analysis.

L 151: Results chapter: Please provide with a graph of androstenone levels versus skatole levels (including all samples), with different marks for the different treatments (T0, T1, T2, T3 & T4)

L 154-156: Please reformulate. Better to talk about numerical reduction, however, non-significant (p= 0.052-0.055). Furthermore, the tendency to decrease skatole with increasing tannins supplementation does not seem to be progressive (table 2 shows no difference between T2, T3 and T4)

L174: A question that comes to my mind is: according to table 2, skatole levels of T0 (0.28 micro-g/g adipose tissue) are above threshold values. Then, how is it that the odour and flavour of these samples seem to be positively evaluated by panelists (above 2.5 in sensory evaluation)? Can you please comment on this?

L171-173, 176: What is the sense of this paragraph? The difference between genders should be accounted for each treatment (in other words, the question of interest would be: do women evaluate the tenderness of T1 samples differently than men?, etc.)

L 176-177: Please discuss about the agreement or not of these results with other authors’ results.

L 193-200: This paragraph does not belong here. Please move it to the introduction chapter.

L 201-202: Please replace “pigs” by “pigs’ adipose tissue”

L205-206: Please reformulate: Your data only shows a tendency to decrease skatole levels, there is no statistical significance.

L 206-210: Please reformulate. Refs. 28 to 30 do not show a significant effect of the supplementation of tannins in the diet, i.e. a decrease in skatole levels in adipose tissue, with respect to the control group. Ref 28 shows a numerical decrease, with the exception of T1 (1% extract supplemented) as you mentioned.

L 212-219: Please reformulate: In agreement with other authors [28-30], the present study does not show any significant effect of tannins feed-supplementation on androstenone levels in adipose tissue. However, ref 28 shows a numerical reduction in androstenone levels in adipose tissue with increasing amounts of diet-supplemented tannins, whereas in the present study an increase of androstenone levels is observed instead.

L 237: Please replace “odour” by “sheep meat odour”

L 246-247: numerical decrease although not significant

L249-251: see comment for L171-173

Author Response

Responds to comments of reviewer 1

At first, due to comments of one of reviewers regarding statistics methods used in our experiment, we recalculated the results. As a result of minor changes that occurred, there were also changed another chapters „Discussion“, Conclusion“, „Abstract“ and „Simple Summary“.

REVIEWER 1

L 11:     Please insert “the risk of” after “to avoid”

L12 – …to avoid the risk of boar…

L 19:     Please consider changing to “2-4% supplementation of tannins in the feed”

L23-24 - … that 2-4% supplementation of tannins in the feed

L57-58: Anosmia to Androstenone was shown to be different according to country/region!

L70-72: This compound has an odour similar to urine or sweat, is perceived by approximately two-thirds of the human population and has been shown to be different according to country/region.  

104-105: “75%” is this figure the hydrolysable tannins content certified by the provider of chestnut extract? Or is it a value determined by you? In any case, please provide the method of analysis.

L115-119: The producer of the Farmatan product is Tanin Sevnica d.d., Sevnica, Slovenia, and the supplier was Product Feed a.s., Luzianky, Slovakia. The content of tannins in this product is 73 ± 2% (the value declared by producer). The analysis of feed was performed according to the Folin-Ciocalteu method [48]. The total phenolic content is expressed as gallic acid equivalents and is 45.1%.

 L 140-145: Please provide a short method description of this analysis and a reference. This is very important as it is the source of your main results together with sensory analysis.

 L163-174: Fat samples (100 g from a part of the belly) from entire males were removed 24 hours after slaughter. One part of each sample was individually packed in a microtene bag, marked and frozen (-20 °C) until panel testing. The second part of each fat sample was transported to the authorized private laboratory of EKOLAB, s.r.o. (Košice, Slovakia), to analyse androstenone and skatole concentrations according to methods of Ampuero Kragten et al. [50] and Bekaert et al. [51]. Briefly, adipose tissue samples were melted in a microwave for 4 min. Liquid fat was transferred to centrifuge tubes (2 mL), and 1.75 mL of extraction solvent methanol:hexane (9:1) was added. The extract was ultrasonically cleaned in a bath at 50 °C for 5 min and centrifuged for 5 min at 10 000 x g. After cooling, approximately 2.0 ml of extract was then injected into a gas chromatograph equipped with an MS detector (Shimadzu GCMS-TQ8030, Shimadzu Corp., Kyoto, Japan) at an injection temperature of 260 °C.

L 151: Results chapter: Please provide with a graph of androstenone levels versus skatole levels (including all samples), with different marks for the different treatments (T0, T1, T2, T3 & T4)

Graph added to manuscript – L204

L 154-156: Please reformulate. Better to talk about numerical reduction, however, non-significant (p= 0.052-0.055). Furthermore, the tendency to decrease skatole with increasing tannins supplementation does not seem to be progressive (table 2 shows no difference between T2, T3 and T4)

Corrected L208-210 - A numerical reduction, however nonsignificant, (P=0.052-0.055) in skatole concentra-tion in fat tissue was found after administration of 2, 3 and 4% tannins compared to the control group.

L174: A question that comes to my mind is: according to table 2, skatole levels of T0 (0.28 micro-g/g adipose tissue) are above threshold values. Then, how is it that the odour and flavour of these samples seem to be positively evaluated by panelists (above 2.5 in sensory evaluation)? Can you please comment on this?

Answer: The level of skatole in T0 is only slightly above the threshold (0.20-0.25 µg/g) and differences between T0 and supplemented groups T1-T4 for odour and flavor are not significant. We think that level of 0.28 was too low for panelists to detect it.

L171-173, 176: What is the sense of this paragraph? The difference between genders should be accounted for each treatment (in other words, the question of interest would be: do women evaluate the tenderness of T1 samples differently than men?, etc.)

Corrected according to new recalculated results:

L228-231: Moreover, significant differences in the evaluation of these two traits were observed between the sexes of the panellists. Women were more critical than men for both tenderness and juiciness traits regardless of diet group (2.70 vs. 3.00, P<0.05, and 2.65 vs. 2.99, P<0.01, respectively).

L 176-177: Please discuss about the agreement or not of these results with other authors’ results.

 L301-308: Contrary to odour and flavour, the other two sensory traits – tenderness and juiciness – in the present study showed significant differences. Panellists, but only men, scored better (P<0.05) tenderness in the control group than in the two supplemented groups (T3-T4), and juiciness in the control group was evaluated better than in T4. This is contrary to the results of a study [30] that reported no significant effect of tannin supplementation on juiciness and/or tenderness. Regarding differences between the sexes of panellists in the present study, women scored tenderness and juiciness worse than men in both the control and supplemented groups.

L 193-200: This paragraph does not belong here. Please move it to the introduction chapter.

L102-105: The main objective of the present study was to assess the impact of hydrolysable tannins on parameters of eating quality considering the possible effect of the sex of consumers as well as to investigate relationships between sensory evaluation and the content of skatole and androstenone in adipose tissue.

L 201-202: Please replace “pigs” by “pigs’ adipose tissue”

L261 - …pig adipose tissue…

L205-206: Please reformulate: Your data only shows a tendency to decrease skatole levels, there is no statistical significance.

L264-265: In the present study, skatole accumulation in adipose tissue tended to decrease with increasing tannin supplementation.

L 206-210: Please reformulate. Refs. 28 to 30 do not show a significant effect of the supplementation of tannins in the diet, i.e. a decrease in skatole levels in adipose tissue, with respect to the control group. Ref 28 shows a numerical decrease, with the exception of T1 (1% extract supplemented) as you mentioned.

L265-268: Similar numerical decreases were observed in other studies after 2-3% [28] or 3% [29] tannin supplementation, but surprisingly, lower supplementation (1.0 or 1.5%) resulted in higher skatole accumulation than in the control group [28,30].  

L 212-219: Please reformulate: In agreement with other authors [28-30], the present study does not show any significant effect of tannins feed-supplementation on androstenone levels in adipose tissue. However, ref 28 shows a numerical reduction in androstenone levels in adipose tissue with increasing amounts of diet-supplemented tannins, whereas in the present study an increase of androstenone levels is observed instead.

L276-279: This result is partially in contrast with that of other studies [28-30] where numerical reductions were found after supplementation of diet with 1-3, 3, and 3% hydrolysable tannin extracts. However, the pigs in the latter study [30] had low levels of androstenone overall.

L 237: Please replace “odour” by “sheep meat odour”

L299-300: …sheep meat odour…

L 246-247: numerical decrease although not significant

L314-315:  In contrast, the concentration of skatole in fat tissue tended to decrease with increasing tannin supplementation.

L249-251: see comment for L171-173

L317-319: The effect of the sex of consumers on sensory evaluation showed significant differences for tenderness and juiciness of pork in favour of men

Reviewer 2 Report

In the manuscript entitled “Eating quality of pork from entire male pigs after supplementation of diet with hydrolysable tannins”, the authors aimed to determine the effect of different doses of tannins (1 to 4% of inclusion in the diet) on the accumulation of skatole and androstenone, which are associated with “boar taint”, and their effects on meat quality. For this purpose, they carried out an experiment with 80 male pigs (65±3.27 kg; 123±4.42 days old), which were distributed, according to a completely randomized design, into five experimental groups: control (absence of tannin), T1 (1% tannin), T2 (2% tannin), T3 (3% tannin), T4 (4% tannin); with 16 animals each, which received the diets for 40 days, until slaughter.

As highlighted by the authors, this manuscript is scientifically important and innovative mainly because, in addition to measuring the hormone content in adipose tissue, they sought to verify the effect of tannin supplementation on pork quality. However, issues related to statistical analysis choice, statistical model, presentation and discussion of results need to be considered when judging the scientific and methodological intuition of this article.

Methods:

1- In the item “Sensory evaluation”, the authors only mention that the scale used in the sensory panel was defined from levels 1 to 5, with 1 being “worst” and 5 the best evaluation. It would be recommended that the authors describe the ordinal classes that formed the sensory panel for each variable;

2- In the “Statistical Analysis”, the authors describe that one way ANOVA was used for all variables, with fixed effects, and that the comparisons between means were performed between groups (T0-T4), using the Bonferroni test with correction for multiple comparisons and that the results were analyzed and presented in terms of mean and standard error of the mean. If one-way ANOVA was used, only the treatment effect is fixed. However, in the results section, the authors address the comparison between genders for the score of all variables associated with the evaluated meat quality. Thus, it would be necessary to present the complete model used in the analyzes and use a two-way ANOVA model. In addition, the authors did not mention: the statistical test used, the level of statistical significance and whether comparisons of treatments were made among themselves or between control and other treatments.

3- Another important point is that the authors opted for a parametric analysis (ANOVA) of the data on an ordinal scale (meat quality scores), which are normally treated with non-parametric analyses. However, it is known that there is no consensus in the statistical and animal science about which analysis is more appropriate in the case of ordinal data. In general, it is recommended to carry out statistical tests, followed by graphical analysis, in order to verify that the assumptions of normality and homogeneity of variance are not being violated. In case of violation, non-parametric analysis or transformation of data to normal distribution is recommended. In this sense, it would be interesting for the authors to comment on what supported the choice of parametric tests.

4- Still on the approach of the results in parametric terms, from the mean score followed by the SEM, it would be important to discuss what the authors think about the fact that the mean of the scores may not represent any of the categories, as in the case of the present work in which the scale used was from 1 to 9 and the values ​​for the different variables measured, are described in mean values, such as ​​2.72 and 2.84.

5- Finally, as a suggestion, the authors could perform statistical correlation analysis (such as Spearman's correlation, between ordinal data or between ordinal and quantitative data) to correlate skatole and andostenone concentration with meat quality variables. Another suggestion would be to carry out a regression analysis to relate the different levels of tannins with the skatole and andostenone concentration in the adipose tissue of pigs.

Results:

6- Tables from 3 to 6 must be reformulated. Note that the central question of the article is not highlighted in the tables: What was mean score for each of the treatments in each evaluated variable? Other questions such as the total sample size and by gender might be added.

7- The perception of organoleptic properties of meat is subject to differences between male and female panellists. In this sense, the authors report the differences between genders for “tenderness”. However, this result was not addressed/discussed in the discussion item and, in the model, it was also not included since the authors describe “one-way” ANOVA.

è Minor comments can be consulted in the PDF of the manuscript attached below, which have been highlighted in red and commented in balloons.

Author Response

Responds to comments of reviewer 2

At first, due to comments of one of reviewers regarding statistics methods used in our experiment, we recalculated the results. As a result of minor changes that occurred, there were also changed another chapters „Discussion“, Conclusion“, „Abstract“ and „Simple Summary“.

All the changes are marked with yellow.

REVIEWER 2

Methods:

  • In the item “Sensory evaluation”, the authors only mention that the scale used in the sensory panel was defined from levels 1 to 5, with 1 being “worst” and 5 the best evaluation. It would be recommended that the authors describe the ordinal classes that formed the sensory panel for each variable

The table 2 (Scale for sensory evaluation of meat) added. – L159

  • In the “Statistical Analysis”, the authors describe that one way ANOVA was used for all variables, with fixed effects, and that the comparisons between means were performed between groups (T0-T4), using the Bonferroni test with correction for multiple comparisons and that the results were analyzed and presented in terms of mean and standard error of the mean. If one-way ANOVA was used, only the treatment effect is fixed. However, in the results section, the authors address the comparison between genders for the score of all variables associated with the evaluated meat quality. Thus, it would be necessary to present the complete model used in the analyzes and use a two-way ANOVA model. In addition, the authors did not mention: the statistical test used, the level of statistical significance and whether comparisons of treatments were made among themselves or between control and other treatments.

Corrected L178-192: The observed data  were evaluated by One Way analysis of variance with fixed effects  by model yij= μ + αi + eij with  N (0,σ2) for androstenone and skatole. For multiple comparisons of treatmenmt means  were used the Bonfferoni tests. The Dunnett test was not used for comparison control to treatment groups. But from the analyses of variance of both traits can be concluded that  differences  between all groups by F test  are nonsignificant.The observations of  senzoric traits were evaluated by the Two way ANOVA with fixed effect, with statistical   model   yijk = μ + αi + (αβ)jj  +   eijk   with  N (0,σ2).“

  • Another important point is that the authors opted for a parametric analysis (ANOVA) of the data on an ordinal scale (meat quality scores), which are normally treated with non-parametric analyses. However, it is known that there is no consensus in the statistical and animal science about which analysis is more appropriate in the case of ordinal data. In general, it is recommended to carry out statistical tests, followed by graphical analysis, in order to verify that the assumptions of normality and homogeneity of variance are not being violated. In case of violation, non-parametric analysis or transformation of data to normal distribution is recommended. In this sense, it would be interesting for the authors to comment on what supported the choice of parametric tests.

The answer of our statistic:

The usage of parametric methods versus nonparametric one is discutabilness. The efficiency of  parametric methods traits is greater than  nonparametric one. Van der Laan nad Verdoonen (1987, Biom. J. 29,6) were analysed problems of usage classical analysis of variance  methods and their nonparametric  counterparts. Toothaker and De Newman (1994, J. Educ. Behav.Stat, 1994) were analysed nonparametric competitors to the  Two-Way ANOVA. By simulation various design (2x2, 2x4 and 4x4) concluded  that the  ANOVA  F suffer from conservative α and power the mixed normal distribution, but is generally recommended, The usage of nonparametric methods in  design and analyses experiments were done by Brunner na Puri (1996, in Handbook of Statistics Vol 13.  Xinye Jiang,  Farr, Fiore and Hu Sun (2018) by the practical experiments with weightloss by various diet of male and female peaples analysed by  car software R, were concluded that the usage of parametric ANOVA is better than  Kruskall Wallis test. Similarly on Nonparametric Two-Way ANOVA was disscused  Holber Ch., 2022 (Internet). Logtransformation data to normality give us the similar results to original one.   The analysed  traits in our experiment were normally distributed by the significance of skewneess and kurtosis for  groups and sexes.  

  • Still on the approach of the results in parametric terms, from the mean score followed by the SEM, it would be important to discuss what the authors think about the fact that the mean of the scores may not represent any of the categories, as in the case of the present work in which the scale used was from 1 to 9 and the values ​​for the different variables measured, are described in mean values, such as ​​2.72 and 2.84.

The answer of our statistic:

The estimated means of analysed traits follow that scores of senzoric traits are points  of the ordinal scale from 1 to 9 (Likert scale).  The better estimation are confidence intervals, by which can be concluded on characteristics of treatments effects and control one.

  • Finally, as a suggestion, the authors could perform statistical correlation analysis (such as Spearman's correlation, between ordinal data or between ordinal and quantitative data) to correlate skatole and andostenone concentration with meat quality variables. Another suggestion would be to carry out a regression analysis to relate the different levels of tannins with the skatole and andostenone concentration in the adipose tissue of pigs.

The answer of our statistic:

The suggestion on analyses  senzoric traits on tanin concentrate of treatments by  the correlation and regression methods will be done in future. The better correlation  estimates than Spearman  one is e.g. polychoric correlation.

Anyway, Spearman correlation coefficients were finally calculated and are presented in the Table 4.

 Results:

  • Tables from 3 to 6 must be reformulated. Note that the central question of the article is not highlighted in the tables: What was mean score for each of the treatments in each evaluated variable? Other questions such as the total sample size and by gender might be added.

The results were recalculated and after that the tables were changed:

L223-231: “Supplementation of the pig diet with tannins did not show any impact on the panellists’ scores for meat odour and flavour (Tables 5 and 6). However, significant differences among the treatment groups were found for tenderness and juiciness (Tables 7 and 8). Men scored both traits significantly better in meat from the control EM group than in meat from the T3-T4 or T4 groups (3.05 vs. 2.74 and 2.81; 3.28 vs. 2.75, P<0.05). Moreover, significant differences in the evaluation of these two traits were observed between the sexes of the panellists. Women were more critical than men for both tenderness and juiciness traits regardless of diet group (2.70 vs. 3.00, P<0.05, and 2.65 vs. 2.99, P<0.01, respectively).”

  • The perception of organoleptic properties of meat is subject to differences between male and female panellists. In this sense, the authors report the differences between genders for “tenderness”. However, this result was not addressed/discussed in the discussion item and, in the model, it was also not included since the authors describe “one-way” ANOVA.

Corrected L186-189: “The observations of sensory traits were evaluated by the Two-way ANOVA analysis with fixed effect using statistical model: yijk = μ + αi + (αβ)jj  +   eijk   with  N (0,σ2).“

 L229-231: Women were more critical than men for both tenderness and juiciness traits regardless of diet group (2.70 vs. 3.00, P<0.05, and 2.65 vs. 2.99, P<0.01, respectively).

Reviewer 3 Report

Dear Authors,

I like this manuscript, it is very simple taking into consideration methods used by you so perhaps it is worth to improve it a little. I have no serious reservations to it.

For me it is obvious that skatole concentration depends on a diet (and age by the way, that’s why in older sows we can detect “boar taint” because of skatole presence), while androstenone concentration depends on hormonal activity. This type of studies is currently needed. Authors forget mentioned (in introduction or discuss)  that boar taint is strongly hereditary and depends on individual predispositions. I recommend do it.

I expect that androstenone concentration was more correlated with individual predisposition than the diet – perhaps it is worth to calculate/check it.

Another thing that occurred to me while reading this text was the possibility of checking the dependencies related to heritability. Perhaps it is possible to check whether some boars were related to each other, they came from the same sire.

In your study you focus only on nutritional influence on boar taint occurrence, what has lower value (but of course important). In summary you shouldn’t forget about genetic aspects.

Author Response

Responds to comments of reviewer 3

At first, due to comments of one of reviewers regarding statistics methods used in our experiment, we recalculated the results. As a result of minor changes that occurred, there were also changed another chapters „Discussion“, Conclusion“, „Abstract“ and „Simple Summary“.

All the changes are marked with yellow.

REVIEWER 3

“R” - For me it is obvious that skatole concentration depends on a diet (and age by the way, that’s why in older sows we can detect “boar taint” because of skatole presence), while androstenone concentration depends on hormonal activity. This type of studies is currently needed. Authors forget mentioned (in introduction or discuss)  that boar taint is strongly hereditary and depends on individual predispositions. I recommend do it.

We agree. The paragraph on this in chapter “Introduction”  was added:

L72-77 - Deposition of androstenone in fat tissue has high heritability estimates (0.55-0.88), which indicates that it is affected mainly by genetics [3-4]. Skatole (3-methyl-indole) is a metabolite derived from microbial catabolism of the amino acid tryptophan in the hindgut of pigs. Its deposition is influenced mainly by environmental factors (h2 = 0.23-0.55).

“R” - I expect that androstenone concentration was more correlated with individual predisposition than the diet – perhaps it is worth to calculate/check it.

We calculated correlations between androstenone and skatole concentrations to sensory traits. Slightly higher values were found among skatole and sensory traits than among androstenone and sensory parameters.

L215-220 – Table 4 was added and text as well: Correlations between observed traits are presented in Table 4. Generally, all the relationships between concentrations of androstenone or skatole in adipose tissue and sensory traits were small and negative. Higher correlations were observed between skatole and eating quality parameters, with the exception of juiciness, which was more correlated with androstenone concentration.

“R” - Another thing that occurred to me while reading this text was the possibility of checking the dependencies related to heritability. Perhaps it is possible to check whether some boars were related to each other, they came from the same sire.

Yes, some of them were related each other. Selection of the litters was random. We have tried to distribute the piglets – entire males from one litter to all used groups, e.i. control and all supplemented. – L110-112

 “R” - In your study you focus only on nutritional influence on boar taint occurrence, what has lower value (but of course important). In summary you shouldn’t forget about genetic aspects.

Text was added to chapter “Abstract”:

L34-37 - Several studies reported heritability estimates for their deposition from medium (skatole) to high magnitudes (androstenone). In addition to efforts to influence boar taint through genetic selection, much attention has also been paid to reducing its incidence using various feeding strategies.
